# The Measurement of Health-Related Quality of Life of Girls with Mild to Moderate Idiopathic Scoliosis—Comparison of ISYQOL versus SRS-22 Questionnaire

**DOI:** 10.3390/jcm10214806

**Published:** 2021-10-20

**Authors:** Edyta Kinel, Krzysztof Korbel, Mateusz Kozinoga, Dariusz Czaprowski, Łukasz Stępniak, Tomasz Kotwicki

**Affiliations:** 1Department of Rehabilitation, University of Medical Sciences in Poznan, 61-545 Poznan, Poland; 2Department of Physiotherapy, University of Medical Sciences in Poznan, 61-545 Poznan, Poland; kkorbel@ump.edu.pl (K.K.); dariusz.czaprowski@interia.pl (D.C.); 3Department of Spine Disorders and Pediatric Orthopedics, University of Medical Sciences in Poznan, 61-545 Poznan, Poland; mkozinoga@hotmail.com (M.K.); lstepniak@ump.edu.pl (Ł.S.); kotwicki@ump.edu.pl (T.K.); 4Department of Health Sciences, Olsztyn University, 10-243 Olsztyn, Poland

**Keywords:** idiopathic scoliosis, health-related quality of life, Italian spine youth quality of life questionnaire, SRS-22

## Abstract

This study aimed to compare the Italian Spine Youth Quality of Life Questionnaire (ISYQOL-PL) versus the Scoliosis Research Society-22 (SRS-22) questionnaire scores evaluating the validity of the concurrent and known-groups. Eighty-one girls (mean age 13.5 ± 1.8 years) with idiopathic scoliosis (IS) with a mean Cobb angle of 31.0 (±10.0) degrees were examined, all treated with a corrective TLSO brace for an average duration of 2.6 (±1.9) years. The patients’ scores were compared as follows: (1) age: ≤13 years vs. >13 years); (2) scoliosis severity: mild (Cobb angle 10–30°) vs. moderate (Cobb angle > 30°); (3) single curve pattern vs. double curve pattern. Lin’s concordance correlation coefficient was used to evaluate the strength of the association between ISYQOL-PL and SRS-22 scores. *t*-tests were applied to assess if the ISYQOL-PL measure and SRS-22 total score were significantly different in the different groups of patients. The concurrent validity analysis showed a moderate correlation (Lin p_ccc_ = 0.47). The ISYQOL-PL showed a significantly better quality of life in mild than moderate scoliosis. The severity of scoliosis but not the age or the curve pattern demonstrated a direct statistically significant effect on patients’ quality of life only when evaluated using the ISYQOL-PL.

## 1. Introduction

Currently, specialists in various medical disciplines include assessing patients’ health-related quality of life (HRQoL) into routine clinical practice. Considering the HRQoL, girls with idiopathic scoliosis (IS) due to visible deformation of the trunk and exposure to long-term stress related to disease and therapy seem to constitute a particular group [1,2]. Their treatment is complex, long-lasting, often occurs in a challenging period in adolescents’ lives, and requires both patients’ and parents’/legal guardians’ approval [3]. The specificity of the 20/24 h brace treatment and its influence on aesthetics, school activities, and sports activities, such as cycling, can significantly affect IS girls’ stress levels’ perception [4,5,6]. In severe scoliosis, an additional stressor may be the fear of possible surgery. In patients after surgery, both mental and physical discomfort due to spinal fusion or a large scar can appear [7]. To summarize, the need to monitor the HRQoL of patients with IS is justified. It allows for, at the right moment, the decision of implementing appropriate psychological therapy to support the treatment. The HRQoL of patients with IS is studied with specific standardized questionnaires. Vasiliadis et al. [8] developed the Brace Questionnaire (BrQ) for adolescents with IS treated by wearing a corrective brace. There are questionnaires that measure the level of stress induced by the deformity (Bad Sobernheim Stress Questionnaire Deformity, BSSQ—Deformity) and the stress caused by the treatment with a brace (Bad Sobernheim Stress Questionnaire Brace, BSSQ—Brace) for patients with IS [9]. The BSSQ questionnaires, however, do not evaluate the overall HRQoL. The Scoliosis Research Society-22 (SRS-22) questionnaire is the most commonly used [10,11,12]; however, new tools are being developed. The Italian Spine Youth Quality of Life (ISYQOL) questionnaire, translated and culturally validated into English, Spanish, and Polish (ISYQOL-PL), is the first questionnaire developed using Rasch analysis to measure the HRQoL in spine deformity during growth [13,14,15,16,17]. Rasch analysis is a statistical technique for evaluating questionnaires and developing questionnaires’ ordinal scores into interval measures [13,14]. According to Caronni et al., Rasch analysis showed that the SRS-22 suffers poor metric properties, which prevents it from adequately measuring patients’ quality of life [13]. Caronni et al. [14] demonstrated the superiority of the ISYQOL over the SRS-22 questionnaire in the Rasch analysis framework. The aim of this study was to compare the performance of the ISYQOL-PL versus the SRS-22 questionnaire (Polish version) to evaluate HRQoL in patients with IS. The comparison was carried out analyzing the concurrent and known-groups validity of both questionnaires. The a priori hypothesis was proposed as follows: The ISYQOL-PL questionnaire provides additional information compared to SRS-22 in evaluating HRQoL in IS girls undergoing conservative treatment.

The results of the present study led to the conclusion that ISYQOL-PL seems to offer advantageous capacities of analysis comparing to the SRS-22 when applied to adolescent girls undergoing non-surgical treatment for mild or moderate idiopathic scoliosis.

## 2. Materials and Methods

### 2.1. Study Population

The sample included eighty-one girls with IS diagnosis treated by the same specialist in orthopedics. The following criteria for inclusion were applied: (1) girls with diagnosed IS at the age of 10–18 years; (2) with a Cobb angle > 10°; (3) under brace treatment (TLSO, at least three months for at least 12h per day); (4) who completed both the ISYQOL-PL and the SRS-22 questionnaires. Exclusion criteria: (1) history of spine surgery; (2) combined spinal deformities (e.g., scoliosis plus spondylolisthesis); (3) history of relevant diseases, surgery, or trauma, including a positive neurologic examination.

All the invited girls with IS selected through the inclusion criteria participated in the study. Table 1 reports the participants’ demographic and clinical data. 

After a brief explanation about the aim of questionnaires, the participants were left to fill in the questionnaires alone, in a separate space, to minimize any influence from parents or medical staff. They completed both questionnaires before clinical evaluation. 

The Institutional Review Board of Poznan University of Medical Sciences approved the study (5 November 2019).

Data from the ISYQOL-PL and SRS-22 were collected from March to May 2020.

Before inclusion in the study, the parents and the patients signed their informed consent. 

### 2.2. Polish Version of the Italian Spine Youth Quality of Life (ISYQOL-PL) Questionnaire

The ISYQOL is a 20 items questionnaire. Each item is scored 0, 1, or 2. Items investigating the presence of spine-related problems (questions 1–4, 7–9, 11–12, and 14–20) are coded 0-1-2 ((0): never; (1): sometimes; (2): often). Conversely, questions 5, 6, 10, and 13, investigating positive reactions, are coded 2-1-0 ((2): never; (1): sometimes; (0): often). The questionnaire provides a total score, with lower scores representing a higher quality of life. The ordinal ISYQOL total score is subsequently converted to an interval measure, expressed on a 0–100% scale, where 100% indicates the highest quality of life. The ISYQOL questionnaire has two domains: Spine health (13 items) and Brace (7 items) specifically devoted to brace wearers. The Rasch method used in the analysis allows for comparing the ISYQOL result of non-brace wearers (who answer only 13 of the 20 items) with brace wearers (who complete the entire questionnaire). In the present study, only the brace wearer IS girls were included. The questionnaire is developmentally appropriate for ages 10–18 years and is designed to be self-administrated [13,14]. The recently validated Polish language version (ISYQOL-PL) of the ISYQOL questionnaire was used in the study [13,17].

### 2.3. Polish Version of the Scoliosis Research Society-22 (SRS-22) Questionnaire

SRS-22 is the criterion standard for measuring the HRQoL in IS. The SRS-22 questionnaire was developed according to the classical test theory and, in this framework, presents satisfactory psychometric properties such as concurrent validity and reliability [11]. It includes 22 items scored on five ordinal categories: Intensity of Pain—5 items (items 1, 2, 8, 11, and 17); Self-image—5 items (items 4, 6, 10, 14, and 19); Function/activity—5 items (items 5, 9, 12, 15, and 18); Mental health—5 items (items 3, 7, 13, 16, and 20); Satisfaction from treatment—2 items (items 21 and 22). The scores for each answer range from 1–5 points. In each category, the recipient can score from 5–25 points, except for the satisfaction from the treatment category, where they can score from 2–10 points. The overall score can range from 22 to 110 points. A total score is obtained by adding individual item scores so that the higher the total score, the better the HRQoL [10,11,12]. A previously validated Polish version of the SRS-22 questionnaire was used in the study [18].

### 2.4. Statistical Analyses

Statistical analysis was applied and managed at various levels beginning with general descriptive statistics for the entire sample. Thus, when applicable, tests for normality were carried out (Shapiro–Wilk Test), followed by tests for homogeneity of variance (Fisher Test).

All statistical analyses were carried out using the Real Statistics Resource Pack software with a significance level of α < 0.05 (Real Statistics Resource Pack software (Release 7.6). Copyright (2013–2021) Charles Zaiontz. www.real-statistics.com) accessed on 17 October 2021.

### 2.5. Known-Groups Validity

There could exist a variety of factors negatively affecting adolescents’ HRQoL. We subdivided patients into different known-groups considering: perception at different ages, pathological severity, and curve pattern.

A measure has high known-groups validity (a type of construct validity) if it discriminates across groups of patients believed to be different on theoretical or clinical grounds [19].

For the known-groups’ validity analysis, the patients’ scores were compared as follows: (1) age: ≤13 years vs. >13 years; (2) scoliosis severity: mild (Cobb angle 10–30°) vs. moderate (Cobb angle > 30°); (3) single curve pattern vs. double curve pattern, Figure 1.

Depending on normality and homogeneity of variance conditions, parametric or non-parametric comparison tests (*t*-test, Mann–Whitney Test, Welch’s Test) were applied to assess if the ISYQOL-PL measure and the SRS-22 total scores were significantly different in the different groups of patients.

### 2.6. Concurrent Validity

The concurrent validity of the ISYQOL-PL was checked using the SRS-22 questionnaire considered the standard measure criterion of HRQol in patients with IS. We used Lin’s concordance correlation coefficient (CCC) as a measure of agreement, which, unlike the intraclass correlation (ICC), does not present the limitation of assuming a common mean for compared ratings at the outset, so it can be used to assess both the level of agreement and the level of disagreement [20,21]. In addition, to compare our results with the reference literature, the Spearman non-parametric correlation test was applied to evaluate the strength of the association between the two questionnaires.

### 2.7. Factors Influencing the HRQoL

Further investigation was carried out to study the influence on the HRQoL given by selected pairs of factors. Such an analysis was performed using a battery of two-way ANOVA tests (α = 5%) examining the following pairs of factors, respectively: (1) age and Cobb angle; (2) number of curves and Cobb angle; (3) years of treatment and Cobb angle.

## 3. Results

### 3.1. Known-Groups Validity

The data were revealed to be normally distributed in both questionnaires, as verified with Shapiro–Wilk test for normality. Therefore, parametric tests were used. The known-groups validity analysis was performed by applying an Independent Two-sample *t*-test. In addition, analysis with a non-parametric approach always confirmed the results obtained with a parametric one.

Both the ISYQOL-PL and the SRS-22 tools showed no difference (1) between the two age groups: ≤13 years vs. >13 years; and (2) between the groups of single vs. double curves. Conversely, the ISYQOL-PL was the only questionnaire showing significantly better HRQoL in mild than moderate scoliosis.

The values for the ISYQOL-PL and the SRS-22 questionnaires for (1) ≤13 years vs. >13 years, (2) Cobb angle 10–30° vs. Cobb angle > 30, and (3) single vs. double curves, are reported in Table 2.

### 3.2. Concurrent Validity

The agreement measured through the Lin’s CCC (p_ccc_ = 0.47) demonstrated a moderate/low concordance. The Spearman correlation coefficient value rho = 0.53, showed a moderate correlation [22] between the two questionnaires.

### 3.3. Factors Influencing the HRQoL

As the data were normally distributed in both questionnaires, we used a two-way ANOVA test. Results confirmed no interactions between: (1) age and Cobb angle; (2) number of curves and Cobb angle; (3) years of treatment and Cobb angle; for the ISYQOL-PL questionnaire, while moderate (<30° Cobb) vs. severe (>30° Cobb) scoliosis is a statistically significant factor influencing HRQoL perception measured through ISYQOL. Conversely, the SRS-22 two-way ANOVA tests highlighted a significant interaction between age and Cobb angle. In particular, the combination of levels of these two factors inhibit each other’s effects, showing a so-called “interference” [23], as shown in Table 3.

## 4. Discussion

The currently published SOSORT (International Scientific Society on Scoliosis Orthopaedic and Rehabilitation Treatment) guidelines for conservative scoliosis treatment promote the HRQoL as one of the essential aims of therapy [24]. This study included girls with IS treated by a corrective TLSO brace, for at least three months for at least 12h per day. It is worth underlining that the treatment’s usefulness has been shown to depend on the patients’ treatment compliance [25,26] and the quality of brace usage [26]. Furthermore, in the recently presented consensus on the best practice guidelines for bracing in adolescent idiopathic scoliosis, some emphasis was put on the patient’s emotional/psychological health as a factor in making bracing decisions [27]. Brace-based treatment, gender, and severity of the disease can significantly interfere with several aspects of patients’ lives, resulting in high levels of stress and a negative impact on daily life [7]. Therefore, using HRQoL questionnaire outcomes is very important in monitoring and assessing the results of treatment.

In this study, we compared for the first time the Polish language version of the ISYQOL-PL with the Polish language version of the SRS-22 questionnaire (considered as a standard criterion for measuring HRQoL in IS). Both questionnaires were culturally adapted and validated as reliable tools for HRQoL evaluation in the Polish language for patients with spinal deformity.

Caronni et al. [14] found that the newly developed ISYQOL questionnaire to measure the HRQoL of adolescents with spinal deformities performs better than the SRS-22 questionnaire when used in the Italian language. The authors claimed that the ISYQOL questionnaire was the first questionnaire developed by using Rasch analysis. Questionnaires built up with Rasch analysis have several advantages compared to questionnaires created according to classical test theory [14]. The same group of authors in a different study using Rasch analysis showed that the SRS-22 questionnaire suffers from poor metric properties, preventing it from adequately measuring patients’ quality of life [28]. Given such limitations, statistical analysis using the SRS-22 should be applied with caution [28]. However, the SRS-22 questionnaire, developed according to the classical test theory, presented satisfactory psychometric properties such as concurrent validity and reliability [11].

Debate is still open on how to statistically treat Likert scale and Likert-type scale data. Indeed, an important and often underestimated limitation of questionnaires comes from their ordinal nature, which does not support proper arithmetic operations assuming linear interval properties (e.g., addition and subtraction) [29,30,31]. Due to the lack of additivity of rating scale data [30], questionnaire scores should be formally excluded from parametric statistics [32].

Some experts view Likert scales as being strictly ordinal in nature; thus, parametric analysis approaches assuming quantitative, or at least interval-level measurements, are not appropriate [33,34,35]. These “ordinalist” views seem to stem from overlooking the difference between individual items and overall Likert scales, as pointed out by those holding the contrasting “intervalist” viewpoint [33,36,37,38].

However, Likert scales can be included in a larger group of measures that are sometimes referred to as summated (or aggregated) rating scales. The reason is because they are based on the idea that some underlying phenomenon can be measured by aggregating an individual’s rating of his/her feelings, attitudes, or perceptions related to a series of individual statements or items [33].

In particular, a restriction should hold for the SRS-22 questionnaire, for which Rasch analysis showed it to suffer from poor metric properties [13]. However, such a position has been criticized as an overly restrictive approach, in practice, since numerous studies have suggested that parametric approaches are acceptable when the data are not strictly on the interval level of measurement [33,39,40,41,42,43].

A very informative review by Harpe on this topic suggests several recommendations to follow in such a case [33]. In particular, in the present study, we considered the first two recommendations when approaching the statistical analysis. Recommendation 1: “Scales that have been developed to be used as a group must be analyzed as a group, and only as a group.” Recommendation 2: “Aggregated rating scales can be treated as continuous data.” Thus, to analyze the ISYQOL-PL and SRS-22 scales’ behaviors (considered as a group), we decided to use parametric statistics [33]. The same approach was applied by Caronni et al. [14], in which they compared the Italian language versions of the ISYQOL and SRS-22 questionnaires using parametric analysis, stating that despite being “aware that parametric statistics should be avoided on ordinal data, however, because it is a common practice to use parametric statistics for SRS-22 analysis, they preferred to compare ISYQOL with the current practice”.

Moreover, in the present study, the non-parametric statistics for the SRS-22 questionnaire confirmed all the findings obtained through parametric statistics.

The known-groups validity analysis demonstrated that both the ISYQOL-PL and the SRS-22 tools showed no difference (1) between the two age groups: ≤13years vs. >13years; and (2) between the groups of single vs. double curves. 

Conversely, the ISYQOL-PL showed significantly better HRQoL in mild rather than moderate scoliosis, as was found by Caronni et al. [14]. This result supports the idea that the ISYQOL-PL questionnaire depicts better what is perceived in clinical observations.

In Caronni et al. [14], the concurrent validity analysis was approached using correlation and the Spearman rho coefficient. Agreement and correlation are widely used concepts in the medical literature. Both are used to indicate the strength of association between variables of interest, but they are conceptually distinct, and, thus, require the use of different statistics [20,21]. In our results, Lin’s CCC p_CCC_ = 0.47 showed low/moderate agreement between the two questionnaires [21]. Additionally, using the Spearman rho = 0.53, the concurrent validity analysis showed a moderate validity [22] of the ISYQOL-PL measure vs. SRS-22. A questionnaire has good concurrent validity if it shows a good agreement with an accepted standard measure [44]. The found correlation is lower than the one found by Caronni et al. for scoliotic patients (Spearman r = 0.71). Conversely, our value appears to be closer to the results (Spearman r = 0.56), which Caronni et al. found for the concurrent validity of the two questionnaires when hyperkyphosis patients were considered [14]. The lack of agreement/correlation could be explained concerning the structure of the questionnaires: SRS-22 is a multidimensional questionnaire (with its five domains) while ISYQOL is unidimensional.

The two-way ANOVA confirmed no interactions between severity of pathology and age, or years of treatment, or number of scoliotic curves. This substantially confirms that only severity of pathology in our sample has a direct statistically significant effect on the HRQoL of patients when evaluated via the ISYQOL-PL tool. Such a result complies with what it is commonly observed in clinical practice. Conversely, the statistically significant interaction (interference) found for SRS-22 shows some limitations of this questionnaire, introducing some unclear outcomes. Indeed, while in the ISYQOL-PL the value of Cobb angle was always a significant factor, showing its importance in the perceived HRQoL level in the SRS-22, the age and Cobb angle values interfere, masking which of the two factors has a direct influence on the perceived HRQoL.

The study’s clinical relevance is related to the need to use tools with high metric properties in daily clinical practice to measure patients’ HRQoL by specialists involved in the treatment process. It is important to underline that the monitoring of the HRQoL is one of the essential recommendations in the treatment of patients with spinal deformities. The ISYQOL-PL will simplify collecting individual results and interpret them adequately and objectively for Polish language clinicians and patients. Health professionals could produce an important practical clinical impact by using such as tool as a basis to discuss with the patient their disease, to support them, clarify common myths and fears associated with the deformity, or help them concerning the brace experience. In addition, the clinician can use the outcomes of the questionnaire to tune the treatment process.

Given that the data collection was during the first wave of the COVID-19 pandemic in Europe, we strongly considered eventual influence impacting the patients’ perceived HRQoL, so we asked the patients to focus on their scoliosis-related issues strictly. The questions refer to spine diseases, and the patients, when answering the questions, understood them in this way. None of the patients presented and reported malaise in connection with the COVID-19 pandemic. In addition, parents or legal guardians did not raise any objections to the behavior and well-being of their children. There were no circumstances that affected the quality of the answers provided.

As a limitation, our study analyzed only girls with IS under brace treatment. Ongoing research aims to compare groups with and without brace treatment, including gender differences and per-spinal deformities differences.

## 5. Conclusions

Both questionnaires adequately measure HRQoL. ISYQOL-PL seems to offer advantageous analytical capacities compared to SRS-22 when applied to adolescent girls undergoing non-surgical treatment for mild or moderate idiopathic scoliosis. The severity of scoliosis but neither the age nor the curve pattern demonstrated a direct statistically significant effect on HRQoL in girls with IS when evaluated using the ISYQOL-PL tool. Such an effect could not be detected using the SRS-22 questionnaire. Using the HRQoL questionnaire, with high metric properties, in different languages, could constitute the basis for creating a multicenter study to better understand the changes in the quality of life of girls with IS during treatment of a larger population with different cultural backgrounds and environments.

## Figures and Tables

**Figure 1 jcm-10-04806-f001:**
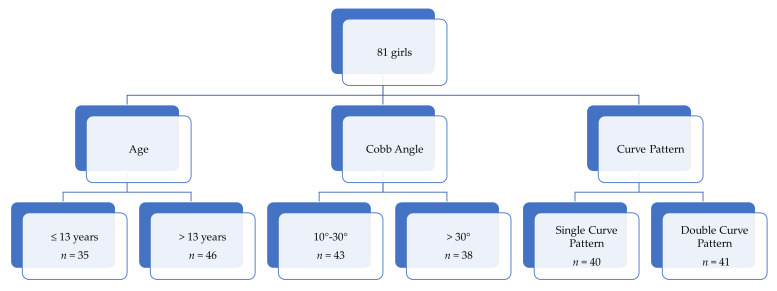
Patients’ distribution according to age, Cobb angle, and curve pattern.

**Table 1 jcm-10-04806-t001:** Participants’ demographic and clinical data.

Number of Participants	Gender	Mean Age (SD) Years	Mean Cobb Angle (SD) Degrees	Mean TLSO Brace Wear (SD) Years
81	females	13.5 (1.8)	31.0° (10.0)	2.6 (1.9)

SD—Standard deviation.

**Table 2 jcm-10-04806-t002:** The known-groups validity analysis of the ISYQOL-PL and the SRS-22 questionnaires.

Questionnaire	Number of Participants	Mean of Score Points (%) (SD)	*p*-Value
ISYQOL-PL ≤13 years vs. >13 years	35 vs. 46	48.1% (6.6%) vs. 49.2% (7.8%)	NS
SRS-22 ≤13 years vs. >13 years	35 vs. 46	48.2% (4.3%) vs. 49.6% (5.3%)	NS
ISYQOL-PL *Cobb angle 10–30°* vs. *Cobb angle > 30°*	43 vs. 38	48.9 (7.4%) vs. 44.8% (8.1%)	0.019
SRS-22 *Cobb angle 10–30°* vs. *Cobb angle > 30°*	43 vs. 38	48.4% (4.6%) vs. 49.6% (5.4%)	NS
ISYQOL-PL *single* vs. *double curves*	41 vs. 40	46.9% (8.6%) vs. 47.0% (7.3%)	NS
SRS-22 *single* vs. *double curves*	41 vs. 40	48.1% (4.7%) vs. 49.9% (5.2%)	NS

SD—Standard deviation; NS—Not Statistically Significant.

**Table 3 jcm-10-04806-t003:** Two-way ANOVA analysis to study the interaction of scoliosis severity level and age and years of treatment on patients’ perceived HRQoL measured with the ISYQOL-PL and the SRS-22 questionnaires.

	Two Factor ANOVA (via Regression) ISYQOL-PL							Two Factor ANOVA (via Regression) SRS-22				
**Interaction Cobb Angle–Age**	**ANOVA**				**Alpha**	**0.050**	**ANOVA**				**Alpha**	**0.050**
		**SS**	**df**	**MS**	**F**	** *p* ** **-Value**		**SS**	**df**	**MS**	**F**	** *p* ** **-Value**
	**Cobb Angle**	342.875	1	342.875	5.578	**0.021 ***	Cobb Angle	39.554	1	39.554	1.674	0.200
	**Age**	0.007	1	0.007	0.000	0.992	Age	19.054	1	19.054	0.807	0.372
	**Interaction**	12.701	1	12.701	0.207	0.651	Interaction	110.054	1	110.054	4.658	**0.034 ***
	**Within**	4733.231	77	61.471			Within	1819.138	77	23.625		
	**Total**	5091.482	80	63.644			Total	1987.777	80	24.847		
**Interaction Cobb Angle–Number of Curves**	**ANOVA**				**Alpha**	**0.050**	**ANOVA**				**Alpha**	**0.050**
		**SS**	**df**	**MS**	**F**	** *p* ** **-Value**		**SS**	**df**	**MS**	**F**	** *p* ** **-Value**
	**Cobb Angle**	342.875	1	342.875	5.578	**0.021 ***	Cobb Angle	33.074	1	33.074	1.349	0.249
	**Number of Curves**	0.007	1	0.007	0.000	0.992	Number of Curves	69.984	1	69.984	2.855	0.095
	**Interaction**	12.701	1	12.701	0.207	0.651	Interaction	2.509	1	2.509	0.102	0.750
	**Within**	4733.231	77	61.471			Within	1887.346	77	24.511		
	**Total**	5091.482	80	63.644			Total	1987.777	80	24.847		
**Interaction Cobb Angle–Years of Treatment**	**ANOVA**				**Alpha**	**0.050**	**ANOVA**				**Alpha**	**0.050**
		**SS**	**df**	**MS**	**F**	** *p* ** **-Value**		**SS**	**df**	**MS**	**F**	** *p* ** **-Value**
	**Cobb Angle**	342.444	1	342.444	5.579	**0.021 ***	Cobb Angle	34.503	1	34.503	1.371	0.245
	**Years of Treatment**	18.904	1	18.904	0.308	0.581	Years of Treatment	13.359	1	13.359	0.531	0.468
	**Interaction**	0.446	1	0.446	0.007	0.932	Interaction	6.211	1	6.211	0.247	0.621
	**Within**	4726.246	77	61.380			Within	1937.882	77	25.167		
	**Total**	5091.482	80	63.644			Total	1987.777	80	24.847		

SS—sum of squares; F—statistic; df—degrees of freedom; MS—mean square; * (bold)—significant values.

## Data Availability

The data presented in this study are available on request from the corresponding author.

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
