# Peer review of "The Measurement of Health-Related Quality of Life of Girls with Mild to Moderate Idiopathic Scoliosis—Comparison of ISYQOL versus SRS-22 Questionnaire"

_jcm, 2021, doi:10.3390/jcm10214806_

Round 1
Reviewer 1 Report
The authors have made changes that have improved study quality.
Author Response
We thank the reviewer for the valuable work and suggestions.
Reviewer 2 Report
Thank you for writing this interesting study to compare the Italian Spine Youth Quality of Life Questionnaire 17 (ISYQOL-PL) versus the Scoliosis Research Society-22 (SRS-22) questionnaire scores evaluating the 18 concurrent and known-groups validity.
After revision, I would recommend some methodological modifications.
Lines 66 to 69 are part of the Material and methods sections. Please, consider rewriting your hypothesis.
Please, remove the conclusion from the introduction section (lines 71-73).
Line 104, please indicate the study code or number given by the Ethical Committee.
Line 106.
Data collection took place from March to May 2020, during the first wave of the Covid-19 pandemic. Do the results of quality of life could be affected by this issue? Could it affect more one type of questionnaire than the other? For instance, SRS22 ask for the past 6-months information and ISYQOL for the current time. Does the temporality of the questionnaires are comparable?
Please, add a paragraph of these in the discussion section.
Line 169. Concurrent validity.
Correct me if I am wrong, but you are trying to demonstrate the agreement between both questionnaires (using SRS22 as the gold standard) using a correlation test. I found here a methodological problem because Pearson or Spearman correlation trends to over-estimate or sub-estimate the agreement between two measures (commonly used continuous data).
I strongly recommend using Lin’s Concordance Correlation coefficient (CCC) or Interclass Correlation coefficient(ICC) to compare the agreement between SRS22 and ISYQOL (Interrater reliability: the kappa statistic (nih.gov)).
Line 179.
How do you plan to identify the factors influencing the HRQoL? Which questionnaire will you use for it? What do you mean by a conditional interaction?
Line 190 to 194. This paragraph is part of the methodology.
Results.
There is not clear for me what we see in the table 1 results.
Accordingly to the description of the questionnaires, SRS22 scores from 22 to 110 points, then punctuation of 44 is equivalent to 50% quality of life. ISYQOL scores from 0 to 100%
Why are the ISYQOL data so low (around 21 points or percentages)?
Please, clarify if the results are points or percentages for each questionnaire.
Which is the interpretation of the results in terms of quality of life for both questionnaires? i.e., for comparison number 1, the 20.7 points of ISYQOL are equivalent/better/worse to the 53.05 points of SRS22 in <13 years?
You find statistical differences in the ISYQOL questionnaire, stratified by Cobb angle. The 3-points differences are clinically meaningful? The interpretation of the results change from 20 to 23 points/percentages?
Line 209. Current validity
Please use an appropriate statistic.
Line 215.
Did you use bivariate comparison for each test?
You must show a table with the results.
Discussion.
The discussion should include the applicability of both questionnaires in clinical practice, not only focusing on other authors' statistical analysis.
A better presentation and interpretation of the results is needed. The author does not have enough information to conclude that one questionnaire is better than the other or gives more information than the other.
Regards
Author Response
We thank Reviewer nr 2 for the valuable work and detailed suggestions.
……………………………………………………………………………………………………………………………………………………
Lines 66 to 69 are part of the Material and methods sections. Please, consider rewriting your Hypothesis.
Please, remove the conclusion from the introduction section (lines 71-73).
…………………………………………………………………………………………………………………………………………………..
Thank you for the comment.
We placed the Hypothesis (lines 67-70) and the main conclusion (lines: 71-73) at the end of the Introduction section following the request of Reviewer nr 1.
Reviewer nr 1 was asking:
"Introduction
It should define the purpose of the work and its significance, mention the main aim of the work and highlight the main conclusions."
We agreed with Reviewer nr 1 because it is with an agreement with JCM Instructions for the authors: "Please highlight controversial and diverging hypotheses when necessary. Finally, briefly mention the main aim of the work and highlight the main conclusions" (https://www.mdpi.com/journal/jcm/instructions)
……………………………………………………………………………………………………………………………………………………
Line 104, please indicate the study code or number given by the Ethical Committee.
……………………………………………………………………………………………………………………………………………………..
Thank you for the indication.
Lines 104-105 – We can't indicate the study code or number given by the Ethical Committee.
Before starting our research, we have presented all descriptions of the study design to our Bioethics Committee at Poznan University of Medical Sciences. The answer we received was as follows: "According to Polish law and GCP regulations, this research does not require approval of the Bioethics Committee at Poznan University of Medical Sciences - it is not a medical experiment. Because of such a conclusion, the Ethical Committee did not assign to the answer any identification code. For such a reason, in our manuscript, we could communicate only the date of the decision of the Bioethics Committee at Poznan University of Medical Sciences.
Additionally, at the request of the Editorial Office, we sent a document with the opinion of the Bioethics Committee at Poznan University of Medical Sciences as one of the attachments with our manuscript.
…………………………………………………………………………………………………………………………………………………………
Line 106.
Data collection took place from March to May 2020, during the first wave of the Covid-19 pandemic. Do the results of quality of life could be affected by this issue? Could it affect more one type of questionnaire than the other? For instance, SRS22 ask for the past 6-months information and ISYQOL for the current time. Does the temporality of the questionnaires are comparable?
Please, add a paragraph of these in the discussion section.
………………………………………………………………………………………………………………………………………………………….
Thank you for the comment:
We added the following paragraph in the discussion section lines: 350-357
"Given that the data collection was during the first wave of the COVID-19 pan-demic in Europe, we strongly considered eventual influence impacting the patients' perceived HRQoL, so we asked the patients to focus on their scoliosis-related issues strictly The questions relate to the reference to the spine diseases, and the patients, when answering the questions, understood them in this way. None of the patients presented and reported malaise in connection with the COVID-19 pandemic. Also, parents or legal guardians did not raise any objections to the behavior and well-being of their children. There were no circumstances that affected the quality of the answers provided."
Line 169. Concurrent validity.
Correct me if I am wrong, but you are trying to demonstrate the agreement between both questionnaires (using SRS22 as the gold standard) using a correlation test. I found here a methodological problem because Pearson or Spearman correlation trends to over-estimate or sub-estimate the agreement between two measures (commonly used continuous data).
I strongly recommend using Lin's Concordance Correlation coefficient (CCC) or Interclass Correlation coefficient(ICC) to compare the agreement between SRS22 and ISYQOL (Interrater reliability: the kappa statistic (nih.gov)).
…………………………………………………………………………………………………………………………………………………………….
Thank you very much for such a specific comment
We do entirely agree with the Reviewer, and in fact, we used Lin's Concordance Correlation coefficient (CCC) (not the ICC because, in general, there is not a linear relationship between the considered variables as requested by the ICC [20,21]). As long discussed in the discussion section, our data derive from aggregated rating scales so they can be treated as continuous data allowing Lin's CCC use. In effect, the Spearman coefficient is related to correlation and the Lin coefficient to agreement. However, we were not including Lin's CCC in the paper because the results presented a moderate agreement between the two rating scales as it was resulting using the Spearman coefficient. We wanted to compare our results with those of the reference study in the literature that used the Spearman correlation for such a reason we did not mention Lin's CCC. In any case, we agree with the Reviewer the Lin's CCC is the appropriate test, so we modified the manuscript introducing the CCC result, leaving the Spearman coefficient for comparison, and we added the following to the Materials and Methods, Results, and Discussion sections.
Materials and Methods section lines: 172-179
"We used Lin's concordance correlation (CCC) as a measure of agreement which, unlike the ICC, does not present the limitation to assume a common mean for com-pared ratings at the outset, so it can be used to assess both the level of agreement and the level of disagreement [20,21]. In addition, to compare our results with the reference literature, the Spearman non-parametric correlation test was applied to evaluate the strength of the association between the two questionnaires."
Results section lines: 215-219
"The agreement measured through the Lin's CCC (pCCC =0.47) demonstrated a moderate/low concordance. The Spearman correlation coefficient value rho=0.53 showed a moderate correlation [22] between the two questionnaires."
Discussion section lines: 313-329
"In Caronni et al. [14] the concurrent validity analysis was approached using correlation and Spearman rho coefficient. Agreement and correlation are widely used concepts in the medical literature. Both are used to indicate the strength of association between variables of interest, but they are conceptually distinct and, thus, require the use of different statistics [20,21]. In our results, Lin's CCC pCCC =0.47 showed low/moderate agreement between the two questionnaires [21]. Also, using the Spearman rho=0.53, the concurrent validity analysis showed a moderate validity [22] of the ISYQOL-PL measure vs. SRS-22. A questionnaire has good concurrent validity if it shows a good agreement with an accepted standard measure [44]. The found correlation is lower than the one found by Caronni et al. for scoliotic patients (Spearman r=0.71). Conversely, our value appears to be closer to the results (Spearman r=0.56), which Caronni et al. found for the concurrent validity of the two questionnaires when hyperkyphosis patients were considered [14]. The lack of agreement/correlation could be explained concerning the structure of the tools: SRS-22 is a multidimensional questionnaire (with its 5 domains), while ISYQOL is uni-dimensional."
- LIU, J.; TANG, W.; CHEN, G.; LU, Y.; FENG, C.; TU, X.M. Correlation and Agreement: Overview and Clarification of Competing Concepts and Measures. Shanghai Arch Psychiatry 28, 115–120, doi:10.11919/j.issn.1002-0829.216045.
- Berchtold, A. Test–Retest: Agreement or Reliability? Methodological Innovations 2016, 9, 205979911667287, doi:10.1177/2059799116672875.
……………………………………………………………………………………………………………………………………………………………
Line 179.
How do you plan to identify the factors influencing the HRQoL? Which questionnaire will you use for it? What do you mean by a conditional interaction?
…………………………………………………………………………………………………………………………………………………………
We thank the Reviewer for the comment.
It was probably a problem of improper use of the English language, so we led the reader to a possible misunderstanding. We did not plan any questionnaire. In our study, we checked if among the patients' characteristics (considering age, the severity of scoliosis, and years of treatment) some of them could have a direct influence on the perceived quality of life so resulting as factors influencing the HRQoL
We removed the term conditional interaction!
……………………………………………………………………………………………………………………………………………………………
Line 190 to 194. This paragraph is part of the methodology.
…………………………………………………………………………………………………………………………………………………………..
Thanks for the comment, but in reality, in the Methods section, we described that parametric or non-parametric tests would be applied to the data depending on the fact the data would normally result or not normally distributed. The data were normally distributed for both the questionnaires, and given such results, we applied the appropriate statistics. This is what is described in lines 195-199, and this is why we put them in the Results section.
Results.
There is not clear for me what we see in the table 1 results.
Accordingly to the description of the questionnaires, SRS22 scores from 22 to 110 points, then punctuation of 44 is equivalent to 50% quality of life. ISYQOL scores from 0 to 100%
Why are the ISYQOL data so low (around 21 points or percentages)?
Please, clarify if the results are points or percentages for each questionnaire.
Which is the interpretation of the results in terms of quality of life for both questionnaires? i.e., for comparison number 1, the 20.7 points of ISYQOL are equivalent/better/worse to the 53.05 points of SRS22 in <13 years?
We Thank Reviewer nr 2 for the detailed questions.
We agree that there could be some confusion, so we changed Table 2 (table 1 of results), including only the values in percentages. To note for the ISYQOL, the score in percentages is derived by Rasch analysis, and they are listed at www.ISYQOL.org website.
You find statistical differences in the ISYQOL questionnaire, stratified by Cobb angle. The 3-points differences are clinically meaningful? The interpretation of the results change from 20 to 23 points/percentages?
…………………………………………………………………………………………………………………………………………………………….
As seen on the above-mentioned www.ISYQOL.org website, the difference from 20 to 23 leads to 5% difference that can be considered clinically meaningful.
Line 209. Current validity
Please use an appropriate statistic.
We removed the Pearson correlation result, introduced Lin's CCC, and left the Spearman correlation result to compare with the reference literature.
Line 215.
Did you use bivariate comparison for each test?
You must show a table with the results.
Thank you for this comment.
We added a table in which all the ANOVA test results are shown for both questionnaires.
--------------------------------------------------------------------------------------------------------------------------------------
Discussion.
The discussion should include the applicability of both questionnaires in clinical practice, not only focusing on other authors' statistical analysis.
A better presentation and interpretation of the results is needed. The author does not have enough information to conclude that one questionnaire is better than the other or gives more information than the other.
……………………………………………………………………………………………………………………………………………………..
Thank you for this comment.
We added the following part at the Discussion section to highlight the implication of the results
lines: 345-349
"The health professionals could produce an important practical clinical impact, using such as tool as a basis to discuss with the patient about their disease to support them, clarify common myths and fears associated with the deformity or help concerning the brace experience. In addition, the clinician can use the outcomes of the questionnaire to tune the treatment process."
In the Conclusion section, we wrote. "Both questionnaires adequately measure HRQoL." In this way, we didn't claim that one is better than the other,based on the statistical results, we just underlined that:
"ISYQOL-PL seems to offer advantageous capacities of analysis comparing to the SRS-22 when applied to adolescents girls undergoing non-surgical treatment for mild or moderate idiopathic scoliosis. The severity of scoliosis but not the age nor the curve pattern demonstrated a direct statistically significant effect on HRQoL in girls with IS when evaluated using the ISYQOL-PL tool. Such effect could not be detected using the SRS-22 questionnaire".
Round 2
Reviewer 2 Report
I want to thank the authors for the improvement of the document. I think it is much better to conclude about the CCC than the correlation test.
Finally, I would like to ask you to remove the collum of significance (yes/no) of table 3 and make a final typos revision.
Thank you again for the effort!
Best regards
Author Response
We thank Reviewer nr 2 for the valuable work and detailed suggestions.
Finally, I would like to ask you to remove the collum of significance (yes/no) of table 3 and make a final typos revision.
Answer:
Thank you for the comment. We removed the significance column (yes/no) of table 3 and made a final typos revision.
This manuscript is a resubmission of an earlier submission. The following is a list of the peer review reports and author responses from that submission.
Round 1
Reviewer 1 Report
This is a well written paper describing the results of health-related quality of life of girls with idiopathic scoliosis
Introduction
It should define the purpose of the work and its significance, mention the main aim of the work and highlight the main conclusions.
Materials and Methods
Add a flow diagram to population and their distribution.
Discussion
Authors should discuss the results and their implications.
Finally, what is the key message from this study?
Is that there is what is the practical impact of this study?